# Respective Correction Rates of Lateral Lumbar Interbody Fusion and Percutaneous Pedicle Screw Fixation for Lumbar Degenerative Spondylolisthesis

**DOI:** 10.3390/medicina58020169

**Published:** 2022-01-23

**Authors:** Norihiro Isogai, Kodai Yoshida, Yuta Shiono, Yutaka Sasao, Haruki Funao, Ken Ishii

**Affiliations:** 1Department of Orthopaedic Surgery, School of Medicine, International University of Health and Welfare (IUHW), Chiba 286-8520, Japan; sasaospine@marianna-u.ac.jp; 2Spine and Spinal Cord Center, Department of Orthopaedic Surgery, International University of Health and Welfare (IUHW) Mita Hospital, Tokyo 108-8329, Japan; blue_note_yokohama@yahoo.co.jp (K.Y.); yuta2001md@gmail.com (Y.S.); 3Department of Orthopaedic Surgery, International University of Health and Welfare (IUHW) Narita Hospital, Chiba 286-8520, Japan

**Keywords:** lumbar degenerative spondylolisthesis, lateral interbody fusion, percutaneous pedicle screw, reduction, indirect decompression

## Abstract

*Background and Objectives:* There are few reports describing the radiographic correction of vertebral slippage in lateral interbody fusion and percutaneous pedicle screw fixation for lumbar degenerative spondylolisthesis. [Objectives] We evaluated the intraoperative surgical correction obtained by lateral interbody fusion and percutaneous pedicle screw procedures. *Materials and Methods:* Fifty patients were included in this study. According to the Meyerding classification, 35 cases were Grade 1 and 15 cases were Grade 2. Mean age was 64.7 ± 6.4 years old. Seventeen cases were male, and 33 cases were female. The mean preoperative % slip was 21.1 ± 7.0%. After lateral interbody fusion, vertebral slippage was corrected using reduction technique by percutaneous pedicle screw. *Results:* The slippage of vertebra was reduced to 11.5 ± 6.5% after lateral interbody fusion procedure and 4.0 ± 6.0% after percutaneous pedicle screw procedure. One year after surgery, the slippage of vertebra was 4.1 ± 6.6%. The correction rate of lateral interbody fusion was 47.7 ± 25.1%, and that of percutaneous pedicle screw was 33.8 ± 2.6%. The total correction rate was 81.5 ± 27.7%. There was no significant loss of correction one year after surgery. The Japanese Orthopaedic Association Score significantly improved from 14.7 ± 4.2 to 27.7 ± 1.7 points at final follow up. No vascular or organ injury was observed during surgery, and there were no postoperative surgical site infections or systemic complications. *Conclusion:* Compared with previous reports, the final correction rate and the correction rate of the percutaneous pedicle screw procedure were particularly high in this study. Lateral interbody fusion and percutaneous pedicle screw using reduction technique provide excellent clinical and radiographic outcomes for patients with lumbar degenerative spondylolisthesis.

## 1. Introduction

Lumbar degenerative spondylolisthesis is a common disease that involves the slippage of the vertebral body and compression of neural structure due to degenerative changes in the adult population which result in mechanical back pain and neuropathy [1,2,3]. The initial treatment for lumbar degenerative spondylolisthesis is non-operative treatment including medications, physical therapy, and bracing. Surgical treatment is indicated for patients with severe neurological deficit or impairment of activities of daily living [1,4,5]. Although several surgical procedures have been reported including decompression, posterolateral fusion, posterior lumbar interbody fusion, or lateral interbody fusion, interbody fusion surgery has some advantages in restoration of the disc height and maintenance of lumbar lordosis compared to decompression or posterolateral fusion [5,6,7,8,9].

Pedicle screw fixation with interbody fusion is the most popular method for the treatment of lumbar degenerative spondylolisthesis to achieve successful bony fusion [5,6,10]. Pedicle screw fixation enables us to tightly stabilize spinal structure and correct spinal deformity including scoliosis and spondylolisthesis [10,11]. However, conventional open pedicle screw placement requires extensive soft tissue and muscle dissection [12,13]. Percutaneous pedicle screw fixation has been developed to avoid soft tissue damage including muscle denervation, atrophy, and pain, and it has become increasingly popular in spinal surgery [14,15]. In combination with the lateral interbody fusion technique, excellent clinical outcomes of indirect neural decompression after lateral interbody fusion and percutaneous pedicle screw fixation have been reported [16,17].

In terms of surgical techniques, the intraoperative reduction of vertebral slippage for lumbar degenerative spondylolisthesis remained controversial. Lian et al. reported that pedicle screws in the slipped vertebra were pulled out during intraoperative reduction [18]. On the other hand, Wegmann et al. reported a correlation between reduction in slippage and good clinical outcome after lumbar spinal fusion [19], and Takahashi et al. reported that appropriate enlargement of the spinal canal was achieved after indirect decompression of lateral interbody fusion and percutaneous pedicle screw reduction even in lumbar degenerative spondylolisthesis patients with severe stenosis [20]. Because indirect decompression is achieved in lateral interbody fusion and percutaneous pedicle screw fixation, the details of reduction should be examined in each reduction process.

Previous reports showed the surgical reduction rate in slippage of lateral interbody fusion and percutaneous pedicle screw fixation [17,21]. However, there are few reports describing the details of the radiographic correction, and the respective correction rates of lateral interbody fusion and percutaneous pedicle screw procedures are still unknown. In this study, we aim to elucidate the surgical correction obtained by lateral interbody fusion and percutaneous pedicle screw fixation procedures for lumbar degenerative spondylolisthesis.

## 2. Materials and Methods

### 2.1. Study Design and Participants

This was a retrospective study at a single institution in Japan. The study was approved by the institutional ethics committee review board (IRB# 5-16-55). Fifty consecutive patients who underwent lateral interbody fusion and percutaneous pedicle screw fixation for lumbar degenerative spondylolisthesis at the L4/5 level between April 2013 and April 2018 were enrolled in this study. Patients with scoliosis whose Cobb angle was more than 10 degrees were excluded from this study. Patients who had previously undergone spinal surgery at the same level were also excluded. Patients’ demographic data, radiographic data, and clinical outcomes were evaluated preoperatively, immediately after surgery, and one year after surgery.

### 2.2. Variables

We collected the following demographic data for each patient: age, sex, height, weight, and body mass index. The radiographic data, including % slip and correction rate, were evaluated using lateral radiographs or intraoperative fluoroscopic images. The respective radiographic correction rates of lateral interbody fusion and percutaneous pedicle screw procedures were evaluated on preoperative and postoperative lateral radiographs and intraoperative fluoroscopic images. In addition, loss of correction was evaluated one year after surgery using lateral radiographs. We also collected sagittal radiographic parameters such as sagittal vertical axis (SVA), lumbar lordosis (LL), pelvic tilt (PT), pelvic incidence (PI), and local lordosis angle of the fused segment on whole-spine standing lateral radiographs, preoperatively and at final follow-up. Bone union and instrument failure were evaluated by a computed tomography (CT) scan one year after surgery. In addition, the correlation coefficient was also analyzed between the respective reduction rates of lateral interbody fusion and percutaneous pedicle screw and preoperative radiographic parameters on whole standing lateral radiographs. Clinical outcomes were evaluated using the Japanese Orthopaedic Association (JOA) scoring system, preoperatively and at final follow-up. Perioperative and postoperative clinical complications were also evaluated.

### 2.3. Surgical Technique

All patients first underwent lateral interbody fusion under general anesthesia. The patient was placed in a true right lateral decubitus position with the left side elevated and taped in this position. Lateral interbody fusion procedures were performed under C-arm fluoroscopic image guidance. All lateral procedures were performed through a 4 cm skin incision. The psoas muscle was separated, and an initial dilator was guided down to the disc. The initial dilator was equipped with an electromyography monitoring system that aided in avoiding the lumbar plexus and genitofemoral nerves. The lateral access retractor was inserted over the final dilator, discectomy and preparation of the endplate were then performed, and the widest interbody device that can be placed safely was inserted.

After the interbody cage placement and skin closure, the patient was placed on a radiolucent operating table in a prone position, and percutaneous pedicle screws were inserted. All steps of pedicle fixation were also performed under C-arm fluoroscopic image guidance. Bilateral, slightly bent rods were inserted, and inferior pedicle screws were tightened. Then, vertebral slippage was corrected using the reduction technique until the superior pedicle screws were tightened into the rods. During reduction, the assistant held the bilateral rod inserters and extended tabs of the inferior percutaneous pedicle screws tightly (Figure 1). After compression and final tightening were performed, the extended tabs of the percutaneous pedicle screws were removed, and wounds were all closed.

### 2.4. Statistical Analysis

Means ± standard deviations were used to describe continuous variables and frequencies, and percentages were used to summarize categorical variables. Baseline patients’ demographics, preoperative scores, and radiographic variables were compared preoperatively, postoperatively, and one year after surgery using an independent *t*-test or Fisher’s exact test, appropriately. A *p*-value < 0.05 was considered statistically significant.

## 3. Results

Fifty patients were included in this study. According to the Meyerding classification, 35 cases were Grade 1 and 15 cases were Grade 2. The mean age was 64.7 ± 6.4 years old. Seventeen cases were male, and 33 cases were female (Table 1). No vascular or organ injury was observed during surgery. The JOA Score significantly improved from 14.7 ± 4.2 to 27.7 ± 1.7 points at final follow up. There were no postoperative surgical site infections or systemic complications. The mean preoperative % slip was 21.1 ± 7.0%. The slippage of vertebra was reduced to 11.5 ± 6.5% after the lateral interbody fusion procedure and 4.0 ± 6.0% after the percutaneous pedicle screw procedure. One year after surgery, the slippage of vertebra was 4.1 ± 6.6% (Figure 2). The correction rate of lateral interbody fusion was 47.7 ± 25.1%, and that of percutaneous pedicle screw was 33.8 ± 2.6%. The total correction rate was 81.5 ± 27.7%. There was no significant loss of correction at one year after surgery. One year after surgery, CTs showed that there were no loosening of pedicle screws or implant failures.

In comparing the sagittal radiographic parameters, there were no differences between preoperatively and at final follow-up in SVA, LL, PT, PI, and local lordosis angle of the fused segment on whole standing lateral radiographs (Table 2).

There were no significant correlations between the preoperative radiographic parameters and the reduction rates of each lateral interbody fusion and percutaneous pedicle screw procedures (Table 3).

### Case Presentation

Fifty-four-year-old woman with lumbar degenerative spondylolisthesis (L4/5) (Figure 3).

From four years ago, she had bilateral sciatica that gradually worsened, and the intermittent claudication was reduced to 3 min. The patient was referred to our hospital. The chief complaint was low back pain and bilateral sciatica at her initial visit, and there was no obvious muscle weakness. The preoperative JOA score was 14 points. Lateral radiograph showed L4 anterolisthesis of Grade 2 (slip ratio of 25%), and magnetic resonance imaging showed severe stenosis at L4/5. Indirect decompression surgery, using lateral interbody fusion and percutaneous pedicle screw was performed. The slippage of L4/5 was corrected from 25% preoperatively to 16% after lateral interbody fusion, and that was reduced to 3% after a percutaneous pedicle screw procedure. One year after surgery, the slippage was 3%. The correction rate of slippage was 36% by lateral interbody fusion and 52% by percutaneous pedicle screw, and the total correction rate was 88%. There was no loss of correction one year after surgery. The final JOA score was 27 points, and there was no neurological complication.

## 4. Discussion

There are a variety of surgical techniques for lumbar degenerative spondylolisthesis including decompression, posterolateral fusion, posterior lumbar interbody fusion, or lateral interbody fusion [6,7,8]. Although the optimal approach for lumbar degenerative spondylolisthesis remains controversial, the rate of fusion and the correction rate of slippage were found to be higher in those who had additional interbody fusion surgery, and several studies preferred interbody fusion surgery [5]. Regarding the reduction procedure of slippage, the correction rate of lateral interbody fusion and pedicle screw procedures for lumbar degenerative spondylolisthesis was reported from 47% to 69.2%, and it is higher than that of posterior interbody fusion [7,17,21]. In addition, in the presence of degenerative changes, it is possible to indirectly decompress the neural foramen and release the facet joint by lateral interbody fusion [4]. In this study, the correction rate of lateral interbody fusion was 47.7%, which was similar to previous reports. This result suggests that lateral interbody fusion is effective to reduce the slippage in patients with lumbar degenerative spondylolisthesis.

In this study, the mean correction rate of percutaneous pedicle screws was 33.8%, and the mean final correction rate was 81.5%, these were particularly high compared to previous reports. In order to obtain sufficient indirect decompression with lateral interbody fusion and percutaneous pedicle screw, a larger slip correction induces a larger expansion of the spinal canal [22]. In patients who underwent posterior lumbar interbody fusion surgery, Heo et al. reported a percutaneous pedicle screw reduction technique with a wide-open rod passing space to reduce the slippage degree by fastening the space, and, subsequently, to achieve a higher correction rate for percutaneous pedicle screws than that of the open method for lumbar degenerative spondylolisthesis [5]. However, the literature on the surgical correction rate for slippage of percutaneous pedicle screws following lateral interbody fusion is scarce. Rigid connection of the pedicle screw and longitudinal rod is needed for the reduction technique, and the rigidity of the construct depends upon the mechanical properties of the implant [23]. In this study, bilateral rod inserters and extended tabs of the inferior percutaneous pedicle screws were held tightly during reduction to increase the rigidity of the connection of the pedicle screw and longitudinal rod. Therefore, this technique might contribute to the high reduction rate of the slippage. In addition, it might be effective to perform this reduction technique successfully using percutaneous pedicle screws with the integrated shape of screw and extended tab, and lateral interbody fusion with the indirect facet release.

There were several reports regarding the disadvantage of the high correction rate of the slippage for lumbar degenerative spondylolisthesis. Lian et al. reported that the pedicle screws in the slipped vertebra were pulled out during intraoperative reduction [18]. Kang et al. reported that a rod reduction device caused screws loosening and correction loss after surgery [24]. However, these studies included the pedicle screw fixation with posterior interbody fusion surgery. In lateral interbody fusion and percutaneous pedicle screw procedures, a large footprint cage of lateral interbody fusion provides strong interbody stability [16], and preservation of the paravertebral muscles with angled percutaneous pedicle screw can maintain spinal stability [5]. In this study, the loss of correction of the slippage was found approximately one year after surgery. Therefore, the high stability of lateral interbody fusion and percutaneous pedicle screws might correlate with lower loss of correction of the slippage. On the other hand, osteoporosis is a crucial risk factor of screw loosening, and bony lateral recess stenosis induces nerve root impingement following screw reduction procedure [22,25]. For patients with osteoporosis, pedicle screw reduction might be contraindicated to avoid screw loosening. For patients with bony lateral stenosis, the position of the superior articular processes at the slipping level should be evaluated using the impingement line, and pedicle screw reduction should be avoided for the patients with an impingement line of grade 2 or 3 [22].

In this study, the standard deviation of correction rate of lateral interbody fusion was larger than that of percutaneous pedicle screw, and there were no significant correlations between the preoperative radiographic parameters and the respective reduction rate of the lateral interbody fusion and percutaneous pedicle screw procedures. Correction rate in lateral interbody fusion is especially unpredictable, and the risk factors of poor reduction of lateral interbody fusion and percutaneous pedicle screw are still unknown. Further analyses are needed to obtain sufficient reduction for slippage.

There were several limitations in this study. First, this was a retrospective study, and the evidence level is inevitably low as a consequence. Second, the indication for surgery and choice of a surgical method and device were determined. Third, there was a lack of control patients who underwent posterior lumbar interbody fusion or open pedicle screw procedures. Therefore, further prospective study should be examined to clarify the correlation between reduction rate and surgical procedure and osteoporosis.

## 5. Conclusions

We evaluated the surgical correction obtained by lateral interbody fusion and percutaneous pedicle screw procedures for lumbar degenerative spondylolisthesis. The correction rate of lateral interbody fusion was 47.7%, and that of percutaneous pedicle screw was 33.8%, and the total correction rate was 81.5%. Compared with previous reports, the final correction rate and the correction rate of the percutaneous pedicle screw procedure were particularly high in this study. Lateral interbody fusion and percutaneous pedicle screw using reduction technique provide excellent clinical and radiographic outcomes for patients with lumbar degenerative spondylolisthesis.

## Figures and Tables

**Figure 1 medicina-58-00169-f001:**
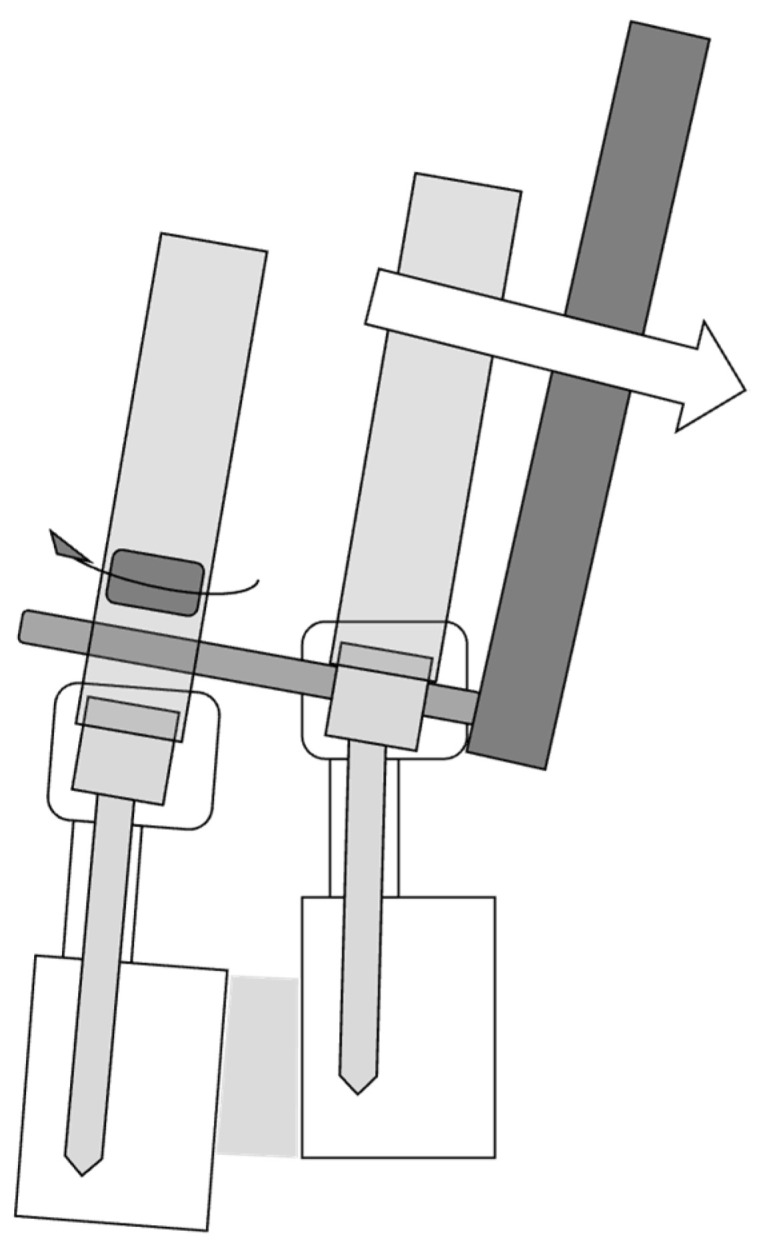
During reduction, the assistant held the bilateral rod inserters and extended tabs of the inferior percutaneous pedicle screws tightly to avoid loose connection between inferior pedicle screws and rods.

**Figure 2 medicina-58-00169-f002:**
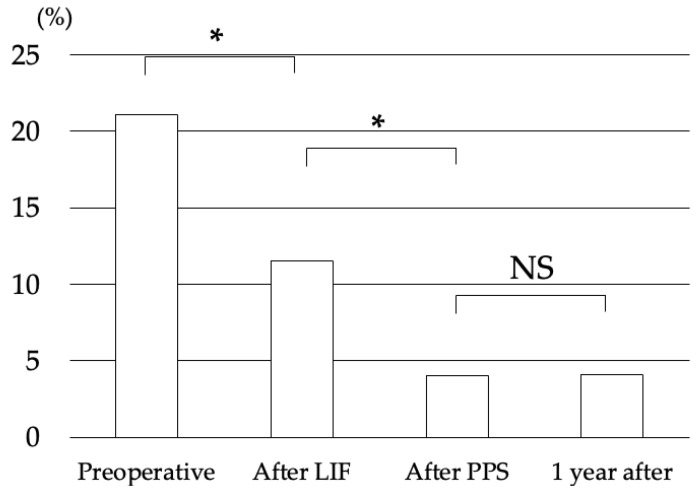
The mean preoperative % slip was 21.1%. The slippage of vertebra was reduced to 11.5% after the lateral interbody fusion procedure, and that was reduced to 4.0% after the percutaneous pedicle screw procedure. One year after surgery, the slippage of vertebra was 4.1%. Abbreviations: LIF, Lateral Interbody Fusion; PPS, Percutaneous Pedicle Screw; NS, Not Significant; *, *p* < 0.05.

**Figure 3 medicina-58-00169-f003:**
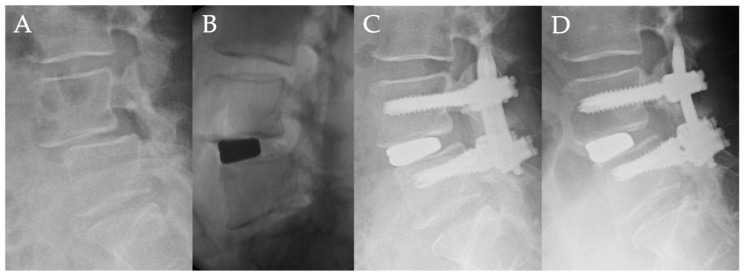
Fifty-four-year-old woman with lumbar degenerative spondylolisthesis: (**A**) Preoperative lateral radiograph showing L4-L5 spondylolisthesis with 25% vertebra slippage. (**B**) Intraoperative fluoroscopic view showed that the slippage of L4/5 was corrected to 16% after lateral interbody fusion. (**C**) Postoperative lateral radiograph showed the slippage of L4/5 was corrected to 3% after percutaneous pedicle screw procedure. (**D**) Lateral radiograph one year after surgery showed no correction loss of slippage.

**Table 1 medicina-58-00169-t001:** Patient’s demographic data.

Age (y/o)	64.7 ± 6.4 (44–90)
Gender	Male 17/Female 33
Height (m)	1.58 ± 0.10 (1.43–1.84)
Weight (kg) BMI (kg/m^2^)	56 ± 11 (36–85) 22.3 ± 3.1 (16.2–30.3)
Meyerding classification Mean preoperative % slip (%) Preoperative JOA score (Pts) Final JOA score (Pts)	Grade 1: 35/Grade 2: 15 21.1 ± 7.0 14.7 ± 4.2 27.7 ± 1.7

Abbreviations: BMI, Body Mass Index; JOA, Japan Orthopaedic Association; Pts, Points.

**Table 2 medicina-58-00169-t002:** Sagittal radiographic parameters on whole-spine standing lateral radiographs preoperatively and at final follow-up.

	Pre-Operation	Final	*p*-Value
Pelvic Incidence (°)	53.1 ± 8.7		
Lumbar Lordosis (°)	40.0 ± 14.3	43.3 ± 12.4	0.41
Pelvic Tilt (°)	25.6 ± 10.9	22.2 ± 11.0	0.30
Sagittal Vertical Axis (mm)	38.2 ± 27.2	39.9 ± 25.3	0.84
Local Lordosis (°)	7.5 ± 4.9	10.0 ± 5.9	0.14

**Table 3 medicina-58-00169-t003:** The correlation coefficient between the reduction rate of lateral interbody fusion and percutaneous pedicle screw and preoperative radiographic parameters on whole spine standing lateral radiographs.

	Lateral Interbody Fusion		Percutaneous Pedicle Screw	
	Correlation Coefficient	*p*-Value	Correlation Coefficient	*p*-Value
Pelvic Incidence	0.04	0.85	−0.39	0.07
Lumbar Lordosis	0.15	0.49	0.03	0.90
Pelvic Tilt	−0.06	0.77	−0.26	0.24
Sagittal Vertical Axis	−0.31	0.17	0.21	0.35
Local Lordosis	0.01	0.97	0.27	0.23

## Data Availability

Not applicable.

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
