# Peer review of "Respective Correction Rates of Lateral Lumbar Interbody Fusion and Percutaneous Pedicle Screw Fixation for Lumbar Degenerative Spondylolisthesis"

_medicina, 2022, doi:10.3390/medicina58020169_

Round 1

Reviewer 1 Report

There is one question - why is there such a large standard deviation of correction rate after LIF procedure (± 25.1%) and the in total correction rate (± 24.7%). Perhaps it makes sense to present the patient data in a table containing the parameters before and after the operation, and it is possible to make some analysis and explanations.

Author Response

We would like to thank Reviewer 1 for the insightful comments and helpful suggestions. The issue raised in this comment is one of the most important points of this procedure. We analyzed the correlation coefficient between the reduction rate of LIF and preoperative radiographic parameters including pelvic incidence, lumbar lordosis, pelvic tilt, sagittal vertical axis, and local lordosis angle at L4/5 to elucidate the risk factors of poor reduction of LIF procedure. However, there was no significant correlation. Therefore, we added the following sentences and table in the material and methods, results, and discussion sections:

Line 159

In addition, the correlation coefficient was also analyzed between the respective re-duction rates of lateral interbody fusion and percutaneous pedicle screw and pre-operative radiographic parameters on whole standing lateral radiographs..

Line 244

There were no significant correlations between the preoperative radiographic parameters and the reduction rates of each lateral interbody fusion and percutaneous pedicle screw procedures (Table 3).

Line 247

Table 3. The correlation coefficient between the reduction rate of lateral interbody fusion and percutaneous pedicle screw and preoperative radiographic parameters on whole spine standing lateral radiographs.

Lateral interbody fusion

Percutaneous pedicle screw

Correlation

coefficient

P-value

Correlation

coefficient

P-value

Pelvic Incidence

0.04

0.85

-0.39

0.07

Lumbar Lordosis

0.15

0.49

0.03

0.90

Pelvic Tilt

-0.06

0.77

-0.26

0.24

Sagittal Vertical Axis

-0.31

0.17

0.21

0.35

Local Lordosis

0.01

0.97

0.27

0.23

Line 378

In this study, the standard deviation of correction rate of lateral interbody fusion was larger than that of percutaneous pedicle screw, and there were no significant correlations between the preoperative radiographic parameters and the respective reduction rate of the lateral interbody fusion and percutaneous pedicle screw procedures. Correction rates in lateral interbody fusion is especially unpredictable, and the risk factors of poor reduction of lateral interbody fusion and percutaneous pedicle screw are still unknown. Further analyses are needed to obtain sufficient reduction for slippage.

Reviewer 2 Report

I conratulate the authors for this paper, I don not have anything to add.

The paper is very well written. The main conclussion for me is that an interbody fusion alone is not suffient inorder to restore the slipage and additionale pedicle srews add significant correction.

Author Response

Thank you for your review and comments.
